# Elevated mutation rates in multi-azole resistant *Aspergillus fumigatus* drive rapid evolution of antifungal resistance

Michael J. Bottery [1] ✉, Norman van Rhijn [1], Harry Chown [1,5], Johanna L. Rhodes [2], Brandi N. Celia-Sanchez [3], Marin T. Brewer [4], Michelle Momany [3], Matthew C. Fisher [5], Christopher G. Knight [6] & Michael J. Bromley [1] ✉

The environmental use of azole fungicides has led to selective sweeps across multiple loci in the *Aspergillus fumigatus* genome causing the rapid global expansion of a genetically distinct cluster of resistant genotypes. Isolates within this cluster are also more likely to be resistant to agricultural antifungals with unrelated modes of action. Here we show that this cluster is not only multi-azole resistant but has increased propensity to develop resistance to next generation antifungals because of variants in the DNA mismatch repair system. A variant in *msh6*-G233A is found almost exclusively within azole resistant isolates harbouring the canonical *cyp51A* azole resistance allelic variant TR$_{34}$/L98H. Naturally occurring isolates with this *msh6* variant display up to 5-times higher rate of mutation, leading to an increased likelihood of evolving resistance to other antifungals. Furthermore, unlike hypermutator strains, the G233A variant conveys no measurable fitness cost and has become globally distributed. Our findings further suggest that resistance to next-generation antifungals is more likely to emerge within organisms that are already multi-azole resistant due to close linkage between TR$_{34}$/L98H and *msh6*-G233A, posing a major problem due to the prospect of dual use of novel antifungals in clinical and agricultural settings.

*Aspergillus fumigatus* is a globally prevalent saprotrophic mould that causes a wide spectrum of diseases in humans. Globally 30 million people are at risk of contracting a life-threatening invasive aspergillosis (IA) infection due to the impact of immunosuppressive medications, neutropenia, and comorbidities such as COPD[1]. Over one million people develop IA annually[2,3], with mortality rates ranging between 30% and 60% even where gold standard treatments are given[4]. In addition, three million people are estimated to have chronic infections caused by *Aspergillus* spp. (chronic pulmonary aspergillosis; CPA) with around 10% succumbing to their infection in the first year and 5-year survival rates of 62%. With a growing at-risk population, these infections are of critical concern resulting in *A. fumigatus* being listed by the WHO as a critical priority fungal pathogen[5,6].

[1]Manchester Fungal Infection Group, Division of Evolution, Infection, and Genomics, Faculty of Biology, Medicine and Health, University of Manchester, Manchester, UK. [2]Department of Medical Microbiology, Radboud University Medical Centre, Nijmegen, Netherlands. [3]Fungal Biology Group and Department of Plant Biology, University of Georgia, Athens, GA 30602, USA. [4]Fungal Biology Group and Department of Plant Pathology, University of Georgia, Athens, GA 30602, USA. [5]Medical Research Council Centre for Global Infectious Disease Analysis, Imperial College London, London, UK. [6]Department of Earth and Environmental Sciences, School of Natural Sciences, Faculty of Science and Engineering, The University of Manchester, Manchester, UK. ✉e-mail: michael.bottery@manchester.ac.uk; mike.bromley@manchester.ac.uk

Resistance to azoles, the primary class of antifungal for the treatment of aspergillosis, is a growing problem, with prevalence reported as high as 10% in some medical centres[7–9]. Mortality rates in those infected with resistant isolates increase by 25% even if alternative salvage therapies are provided[10–13]. The evolution of azole resistance during treatment of patients with chronic disease is typically associated with the acquisition of specific de novo non-synonymous mutations in *cyp51A*[14], leading to conformational changes in the target of azoles—sterol-14α-demethylase—which catalyses a critical step in the synthesis of ergosterol, an essential component of the fungal cell membrane. There is also compelling evidence that the widespread use of agricultural fungicides functionally analogous to clinical azoles select for pan-azole resistance in the environment resulting in a hall-mark set of mutations in *cyp51A*[15]. Environmental pan-azole resistance is often characterised by a 34-base pair tandem repeat in the promoter of *cyp51A* that leads to the over-expression of the gene, coupled with point mutations in *cyp51A* (notably the substitution of leucine 98 with histidine) which reduce the binding affinity of the azoles[16]. The first isolates with $TR_{34}/L98H$[17], which provide high levels of resistance to itraconazole and voriconazole, were observed in the Netherlands in both environmental and clinical isolates, prompting the hypothesis that environmental selection is driving resistance[18]. Subsequently, isolates with these mutations and similar sets of mutations (e.g. $TR_{46}/Y121F/T289A$[19]) have been identified globally[15]. Genome sequencing and molecular epidemiology have now identified multiple examples of near-identical genotypes from both environmental and clinical sources[20], providing strong evidence for patient acquisition of azole-resistant *A. fumigatus* from environmental sources.

Despite the fact that *A. fumigatus* can undergo a sexual cycle and can also readily disperse on air currents due to the production of aerosolised conidia during asexual reproduction, recent genomic studies have identified strong population structuring linked to the presence of pan-azole resistance mutations[20,21]. Phylogenetic analyses of *A. fumigatus* isolates from the United Kingdom showed pronounced genetic clustering into two populations, clade A and clade B, with the azole resistance mutation $TR_{34}/L98H$ almost uniquely occurring in clade A and wild-type *cyp51A* predominantly occurring in clade B[20]. A third divergent azole-sensitive clade has also been identified within global isolate collections[21]. This population genetic structure is reinforced by low levels of recombination between clades and high rates of recombination within clades[21]. Moreover, the genomes in clade A show the signatures of strong directional selection across multiple loci known to be involved in resistance to clinical and agricultural fungicides[20]. Intriguingly, several other variant loci, not currently associated with resistance to agricultural fungicides are also enriched in this clade. Whether these loci have become enriched through hitchhiking with beneficial resistance mutations that are undergoing a selective sweep or are playing a direct role in the evolution of the clade remains unclear. The near uniform association between resistance across multiple antifungal classes[22] and strains in clade A led us to hypothesise that this clade was evolving more rapidly in response to the antifungal challenge, perhaps driven by the impact of these variants.

Defects in DNA mismatch repair (MMR) can result in elevated mutation rates in bacterial, fungal and mammalian cells[23]. Elevated mutation rates in mutator microorganisms can confer an evolutionary advantage when adapting to novel, stressful or fluctuating environments but are often associated with a fitness cost in stable environments due to the accumulation of deleterious mutations. The frequency of mutator strains within a population can be rapidly amplified through genetic hitchhiking upon beneficial adaptive mutations and can play an important role in the evolution of pathogens to novel stresses[24–26]. Mutator phenotypes are key drivers of within-host evolution in response to the host immune system, abiotic stress, and antibiotic treatment, particularly during long-term chronic infection. Elevated mutation rates have been identified in human fungal pathogens, including *Candida albicans*[27], *Nakaseomyces glabrata* (previously named *Candida glabrata*)[28], *Cryptococcus deuterogattii*[29,30] and *Cryptococcus neoformans*[31,32], and may play a role in the acquisition of resistance in *A. fumigatus*[33]. Critically, elevated mutation rates are likely to influence the ability to adapt to strong fluctuating directional selection within the environment, for example, through the use of multiple different fungicides. We therefore investigated if polymorphisms in the MMR system were enriched in the multi-azole-resistant clade A.

## Results

### Variants in mismatch repair genes are overrepresented in clade A isolates with $TR_{34}/L98H$

Eukaryotic MMR consists of two major recognition complexes: MutSα (Msh2-Msh6), which recognises base–base mismatches and small loops, and MutSβ (Msh2-Msh3), which recognises larger loops, with a bias towards deletion loops[34]. The MutLα protein complex (Pms1-Mlh1) directs downstream protein–protein interactions, is required for daughter strand recognition, and has endonuclease activity. We screened 218 previously sequenced *A. fumigatus* isolates[20] 65 originating from environmental and 153 from clinical sources in the United Kingdom (Supplemental dataset 1), of which 91 contain $TR_{34}/L98H$ and 7 contain $TR_{46}/Y121F/T289A$ *cyp51A* azole resistance mutations, for variants in MMR genes *msh2* (AFUB_039320, AFUA_3G09850), *msh3* (AFUB_090020, AFUA_7G04480), *msh6* (AFUB_065410, AFUA_4G08300), *pms1* (AFUB_029050, AFUA_2G13410) and *mlh1* (AFUB_059270, AFUA_5G11700). In total, across all five genes, 212 non-synonymous point mutations were present relative to the *Af*293 reference strain (Supplemental dataset 1). No frameshift, nonsense, or truncation mutations were observed in any of the isolates; thus, variants were expected to alter rather than abolish the activity of the MMR systems. Of these variants, non-synonymous mutations in *msh2* (c.A2435G, p.E812G), *msh6* (c.G698C, p.G233A) and *pms1* (c.A1331G,p.E444G) were significantly associated with clade A (Fig. 1a–c, *msh2*: d.f. = 1, $\chi^2 = 8.88$, $P = 0.0029$, *msh6*: d.f. = 1, $\chi^2 = 141.64$, $P < 2.2 \times 10^{-16}$, *pms1*: d.f. = 1, $\chi^2 = 19.12$, $P = 1.2 \times 10^{-5}$); however, only the G233A variant in *msh6*, which occurs prior to the annotated N-terminal MutS domain responsible for mismatch recognition (Fig. 1e), was significantly associated with the presence of the azole resistance mutation $TR_{34}/L98H$ in *cyp51A* (d.f. = 1, $\chi^2 = 122.27$, $P < 2.2 \times 10^{-16}$). With the exception of the variants we have identified within clade A of *A. fumigatus*, the G233 amino acid is perfectly conserved across over 150 million years of evolutionary history[35] within the *Trichocomaceae* family (Fig. 1f). In total, 85% (105/123) of clade A isolates contained the G233A variant allele of *msh6*, while the variant is only present in 3% of isolates in clade B (3/95). Of the 86 $TR_{34}/L98H$ azole-resistant genotypes, 96.5% contained G233A, whereas. of the 7 isolates containing the $TR_{46}/Y121F/T289A$ resistance haplotype, only 4 harbour the *msh6*-G233A variant (Fig. 1d). Notably, *msh6* is encoded on chromosome 4 0.36 Mbs away from the *cyp51A* gene, however, previous population genomic studies have shown that genetic linkage can decay rapidly even within the resistant cluster[21]. The *cyp51A* gene has a high average fixation index ($F_{ST}$) of 0.1127 (standard error (se) = 0.025) between isolates within clades A and B, implying population subdivision at this locus (average genome-wide $F_{ST} = 0.086$, se = 0.00026). In comparison, *msh6* also has a high average $F_{ST}$ of 0.1386 (se = 0.0422). A two-tailed *t*-test assuming unequal variances between chromosome 4 (where both *msh6* and *cyp51A* are located) and the whole of the genome recovers a significant *p*-value of $1.367^{e-67}$, suggesting multiple loci across chromosome 4 are associated with azole drug resistance. In addition, previous analysis shows this region has a significant association with itraconazole resistance[20] (treeWAS $P < 0.001$). The association between G233A, clade A and $TR_{34}/L98H$ was also evident in a global collection of isolates[36] (Fig. S1, Clade A: d.f. = 2, $\chi^2 = 463.93$,

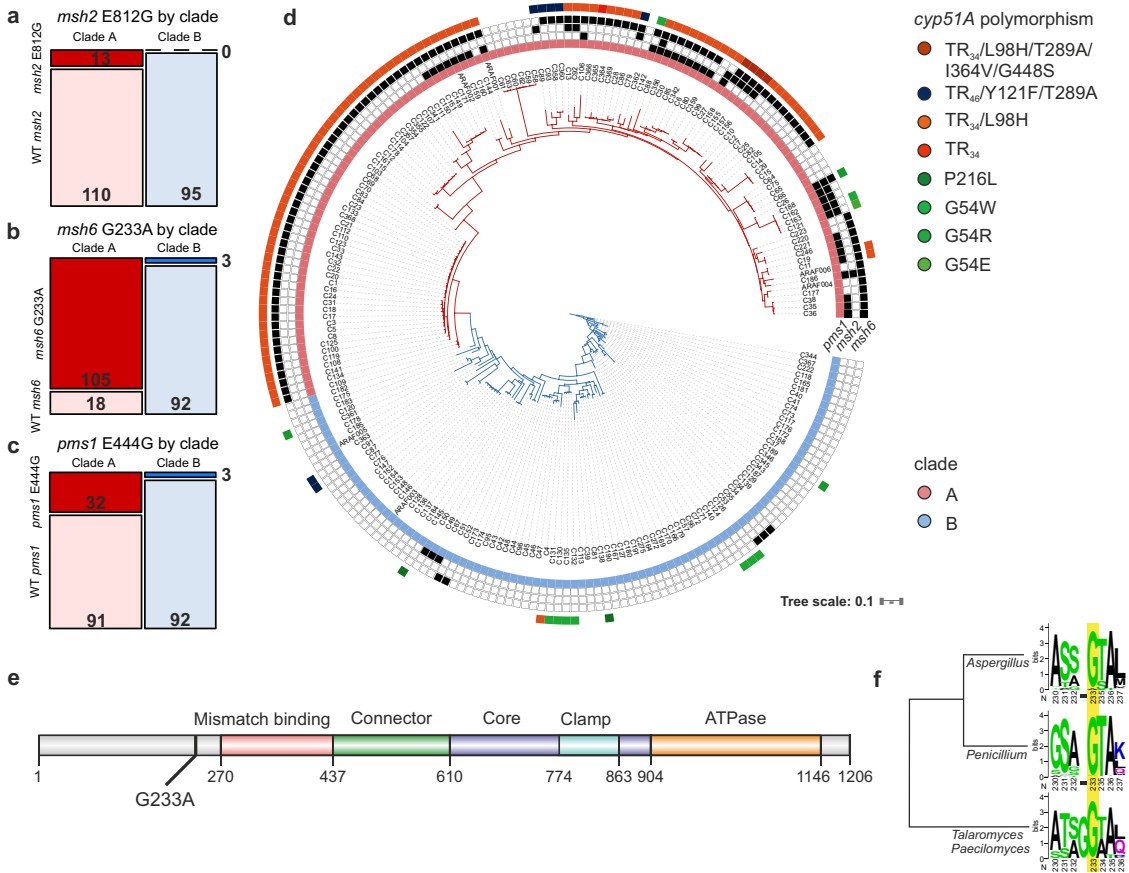

**Fig. 1 | Variants in MMR are significantly overrepresented in clade A.** Occurrence of MMR variant alleles in clade A (red) and clade B (blue) for **a** *msh2* (E812G), **b** *msh6* (G233A) and **c** *pms1* (E444G). **d** An unrooted maximum-likelihood phylogenetic tree using genome-wide SNPs relative to *Af*293 of 218 WGS UK isolates[20]. The presence of variants in *msh2*, *msh6* and *pms1* are highlighted in the black boxes, the presence of *cyp51A* resistance variants and clade are coloured. **e** Domain structures of Msh6 in *A. fumigatus*, G233A variant labelled, labels show predicted domain positions in protein sequence. **f** G233 locus homology across Trichocomaceae, alignments of 140 isolates spanning *Aspergillus* ssp., 27 *Talaromyces* and *Paecilomyces*, and 122 *Penicillium* Msh6 protein sequences. G233 highlighted in yellow, G233A variants are only present in *Aspergillus fumigatus*. Cladogram shows the hierarchical clustering of Msh6 protein sequences. Source data are provided as a Source Data file.

$P < 2.2 \times 10^{-16}$, TR$_{34}$/L98H: d.f. = 1, $\chi^2 = 326.23$, $P < 2.2 \times 10^{-16}$). Thus, the presence of the non-synonymous variant G233A in *msh6*, which is an essential component of MutSα, responsible for recognising base–base mispairing, is strongly associated with the presence of azole resistance allele TR$_{34}$/L98H in clade A.

## MutS and MutL null mutants result in a hypermutator phenotype

Given variant alleles in *msh2*, *msh6* and *pms1* are significantly associated with clade A, and in the case of *msh6* TR$_{34}$/L98H azole resistance genotypes, we first asked whether each of the three genes influence mutation rate. The three genes were independently deleted from the wild-type strain MFIG001, a laboratory strain that clusters within clade B[37]. The minimal inhibitory concentrations to voriconazole, a current antifungal used to treat *A. fumigatus* infections, or the phase III clinical trial compound olorofim in the orotomide class[38] were not altered in the MMR defective strains relative to the parental MFIG001 strain (Fig. S2) indicating no direct effect of these alleles on azole or orotomide sensitivity. To measure mutation rates, a modified Luria–Delbrück fluctuation test[39] was implemented in which mononucleated spores from replicate cultures grown without selection were challenged with voriconazole to determine the probability that spores would spontaneously gain mutations that provide resistance (see the "Methods" section and Fig. 2a). The rate of spontaneous mutation in the wild-type MFIG001 strain to voriconazole was $2.78 \times 10^{-10}$

($\pm 6.9 \times 10^{-11}$) per spore, similar to rates measured in other fungal species[40–42]. The modified Luria–Delbrück method was validated by treatments with the mutagen ethyl methanesulfonate during growth. The method detected the linear increase in mutation rate in MFIG001 to voriconazole with increasing concentrations of the mutagen (linear regression, $R^2 = 0.92$, $F_{1,13} = 153$, $P < 0.001$, Fig. S3). Mutation rates for voriconazole resistance in the MMR deletion strains Δ*msh2*, Δ*msh6* and Δ*pms1* were ~85-, ~47- and ~173-fold higher than the parental wild-type strain (Fig. 2b, Two sample ML-test, MFIG001-Δ*msh2* $T = -3.83$, $P < 0.001$, MFIG001-Δ*msh6* $T = -3.82$, $P < 0.001$, MFIG001-Δ*pms1* $T = -4.12$, $P < 0.0001$). The MICs of spontaneous resistant mutants to voriconazole ranged from 4 to 32 µg/ml and were not dependent upon the genetic background (Kruskal–Wallis, d.f. = 2, $\chi^2 = 4.25$, $P = 0.119$). Sequencing of the *cyp51A* gene showed that 37% (10/27) of the randomly selected spontaneously resistant isolates had the known voriconazole resistance allelic variant G448S (4/8 MFIG001, 2/7 Δ*msh2*, 3/6 Δ*msh6*, 1/6 Δ*pms1*), but these exclusively occurred within isolates with MICs ≥ 16 µg/ml. No mutations within *cyp51A* were observed in the other resistant isolates and tandem repeats in the promotor of *cyp51A* were never observed.

Whole genome sequencing (WGS) of a further five randomly selected spontaneous voriconazole-resistant isolates from the wild-type MFIG001 strain revealed that only one mutational event was detectable in the entire genome of each resistant strain, all generating a G448S variant in *cyp51A*. In contrast, sequencing of five resistant

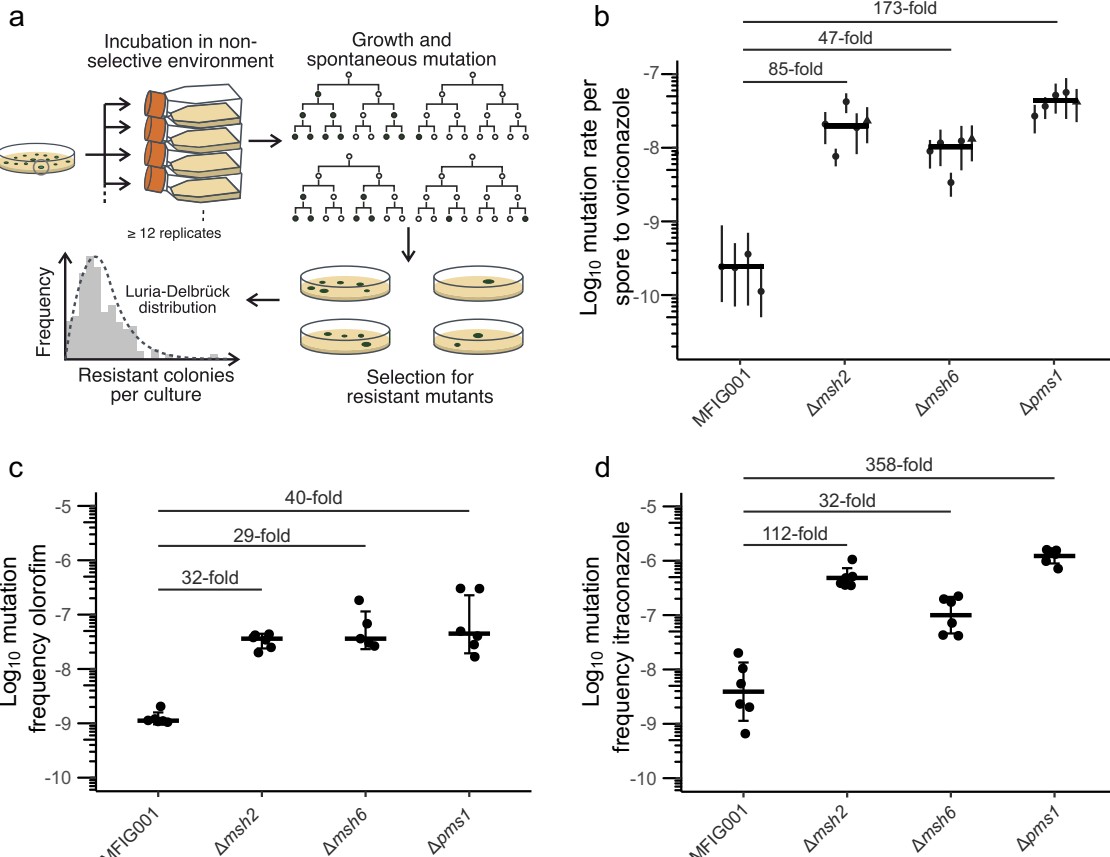

**Fig. 2 | MutS and MutL null mutants significantly elevate mutation rates.**
**a** Workflow of fluctuation tests to measure mutation rates in *A. fumigatus*. Clonal isolates were cultured in the absence of antifungal selection to generate genetic diversity. Resistant mutants were selected on lethal concentrations of antifungals. Counts of resistant mutants were fitted to the Luria–Delbrück distribution to calculate the number of mutational events. **b** Mutation rates for resistance to voriconazole for MMR-deficient mutants. Each point shows the calculated mutation rate from a single independent fluctuation test using 12 replicate cultures. Error bars show 95% confidence intervals, cross bars show the median mutation rate

across fluctuation tests. Fold differences show median fold change in mutation rate from the parental MFIG001 strain. Triangle points represent mutation rates measured using independently constructed deletion mutants. **c** Mutational frequency to olorofim resistance. **d** Mutational frequency to itraconazole resistance. Each point shows the mutational frequency of an individual population ($N = 6$). Error bars show SEM and cross bars show median mutational frequency. Fold differences show median fold change in mutation frequency from the parental MFIG001 strain. Source data are provided as a Source Data file.

isolates derived from the *msh2* null mutant and 6 from the *msh6* and *pms1* null mutants resulted in 28 (s.e. 4), 48 (s.e. 6), and 35 (s.e. 3.2) mutations (synonymous, non-synonymous and intergenic) per genome respectively (Fig. S4a), including canonical azole resistance mutations *cyp51A* G448S (2/5 Δ*msh2*, 3/6 Δ*msh6*, 2/6 Δ*pms1*) and HMG-CoA s in 2/5 Δ*msh2* resistant mutants (Supplemental dataset 2). The deletion of *msh2* resulted in higher frequencies of transversions relative to the deletion of *msh6* or *pms1* (Fig. S4b, Tukey multiple comparisons, Δ*msh6*-Δ*msh2* $P < 0.05$, Δ*pms1*-Δ*msh2* $P < 0.05$) mirroring previously published results showing that the loss of function of Msh2 results in a bias towards transversions[43]. The specific base substitutions showed that the bias towards transversions in the Δ*msh2* strain was due to an over-representation of C>A mutations (Fig. S5a), however there were no clear trinucleotide signatures that were associated with the transversions (Fig. S5b). The proportion of intergenic mutations also differed significantly between deletion mutants (ANOVA, $F_{2,13} = 33.37$, $P < 0.001$, Fig. S4c). While the deletion of *msh6* resulted in a similar ratio of intergenic mutations (Δ*msh6* 48.4% intergenic, s.e. 2.23) to the proportion of intergenic regions in the *A. fumigatus* genome (48.76%), the Δ*pms1* and Δ*msh2* mutations were overrepresented by intergenic mutations (Δ*pms1* 76.2% intergenic, s.e. 2.45, Δ*msh2* 59.8% intergenic, s.e. 2.68) suggesting that the elevated mutation rates within these strains result in deleterious or lethal intragenic mutations

which are purged by negative selection, and suggesting that a high fitness costs may be associated with the deletion of these genes. Mutations in *msh2* have previously been associated with elevated mutation rates and the acquisition of antifungal resistance in *Cryptococcus deuterogattii*[30] with an overrepresentation of mutation occurring with homopolymer nucleotide runs[29]. WGS data from the MMR null mutants showed that while single nucleotide variants did not occur within homopolymer runs (Fig. S6a), single base-pair indels were strongly associated with homopolymer nucleotide runs (Fig. S6b), with indel events occurring in homopolymer runs with a mean length of 9 base pairs. However, indels were not clearly associated with resistance, with no mutations showing parallelism between independent resistant mutants. Moreover, only 11% of indel events occurred within protein-coding regions, despite 49% of the *A. fumigatus* A1163 genome being protein-coding.

The frequency of resistance to another azole-class antifungal, itraconazole and the dihydroorotate dehydrogenase (DHODH) inhibitor olorofim showed similar significant increases in mutation rate in all three MMR deficient strains, with the largest increases in Δ*pms1* and the lowest increases in mutation rate in Δ*msh6* (pairwise Wilcoxon tests, $P < 0.05$, Fig. 2c, d). The probability of resistance arising differed between antifungals (ANOVA, $F_{2,225} = 42.49$, $P < 0.001$), with resistance arising between 2 and 15 times more frequently to itraconazole than

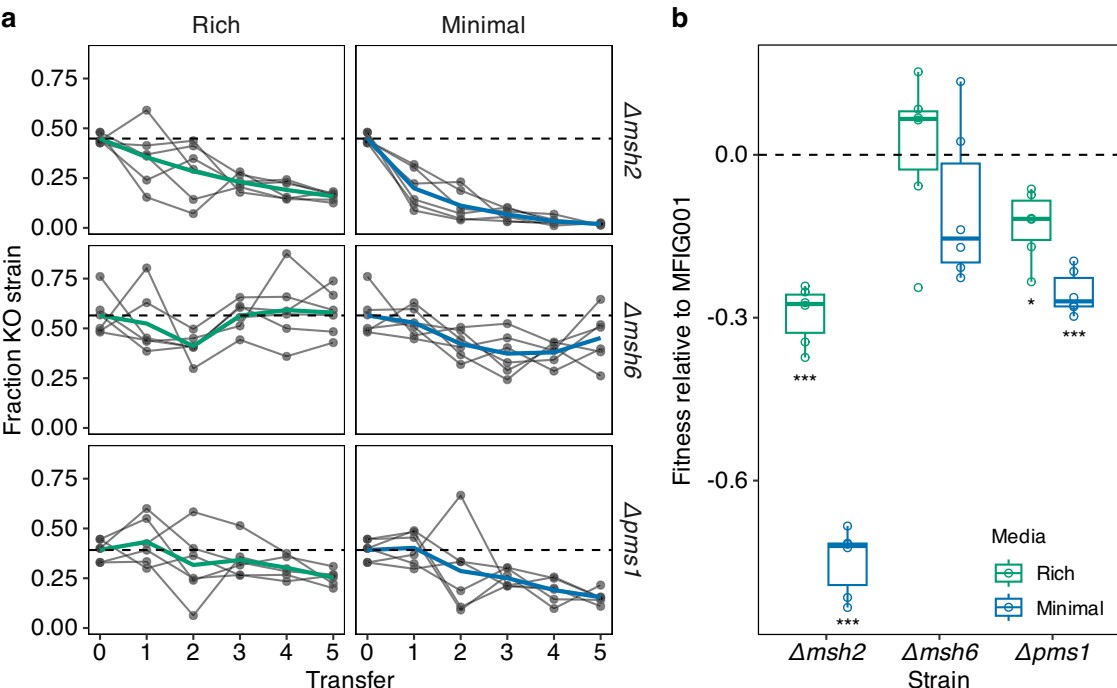

**Fig. 3 | Deletion of MMR genes carries a significant cost to fitness. a** The fraction of *Δmsh2, Δmsh6* and *Δpms1* through time when in direct competition with the parental MFIG001 strain on solid agar faceted by media type (rich = aspergillus complete media, minimal = aspergillus minimal media). The coloured lines show the mean of 6 independent competitions, presented by individual grey lines. The horizontal dashed lines show the starting fraction of the MMR deletion strain. **b** Mean fitness of *Δmsh2, Δmsh6* and *Δpms1* relative to MFIG001 across the five transfers presented in panel a, the cross bar shows the median (*N* = 6 independent

competitions), the lower and upper hinges correspond to the first and third quartiles and the whiskers extend to 1.5*IQR. Box plots coloured by media type. The horizontal dashed line shows equal fitness of zero, *p*-values show significant difference from zero using two-sided *t*-tests, using Holm correction for multiple testing, *Δmsh2* rich *P* = 0.00023, minimal *P* = 4.8e$^{-6}$, *Δmsh6* rich *P* = 1, minimal *P* = 0.97, *Δpms1* rich *P* = 0.025, minimal *P* = 0.00012, asterisks represent significance (*$p < 0.05$, **$p < 0.01$, ***$p < 0.0001$). Source data are provided as a Source Data file.

olorofim or voriconazole (Tukey post hoc test, $P < 0.001$), however, there was no difference between the frequency of resistant mutants to voriconazole and olorofim (Tukey post hoc test, $P = 0.828$). Of the randomly selected olorofim spontaneous resistant mutants, 28/28 had mutations in the *pyrE* resistance hotspot G119[44,45], which provided high levels of olorofim resistance (>2 μg/ml). Together, these results show that *msh2* and *pms1* null mutants, which abolish the activity of the MMR system, result in highly elevated mutation rates that facilitate the emergence of resistance. Moreover, although MutSβ recognition complexes remain functional when disrupting *msh6*[34], the loss of function of *msh6* still results in significant increases in mutation rate.

**Defective MMR results in significant reduction in fitness**

Uncontrolled mutation can result in the accumulation of deleterious mutations, which decrease the fitness of hypermutator strains within stable environments[46,47]. We therefore asked whether such costs were associated with the MMR defects in *A. fumigatus*. Though we detected non-synonymous variants in *msh2, msh6* and *pms1* no predicted loss of function mutations within the MMR genes were observed in the clinical or environmental isolates sequenced, suggesting that the complete loss of MMR could be associated with a significant cost in *A. fumigatus*. However, no defect in radial growth rates was observed over 96 h in the MMR deletion mutants relative to their parental strain in either nutrient-rich or minimal growth conditions (Fig. S7). Interestingly, morphological sectoring, likely to occur due to mutations during hyphal growth, occurred within the MMR mutants but not the parental strain. To determine whether fitness costs manifested over longer periods of growth, MMR-deficient mutants directly competed with the parental strain over five serial transfers on complete (rich) and minimal

solid media (Fig. 3) in the absence of antifungal selection. The frequency of MMR deleted strains decreased through time and was dependent upon both the MMR mutant and the environment (Mixed effects linear model, Transfer:Media $F_1 = 7.8447$, $P < 0.01$, Transfer:Strain $F_2 = 7.8528$, $P < 0.001$, Fig. 3a). Growth medium had a significant effect on the relative fitness in all MMR deficient strains, with fitness being consistently lower in minimal media compared to rich media (Wilcoxon test, $W = 233$, $P < 0.05$, Fig. 3b). Deletion of *msh2* resulted in the highest overall cost relative to the parental strain, displaying 30% cost in rich media (*T*.test, $T_5 = -13.6$, $P < 0.001$, Holm adjusted for multiple testing) and a 75% cost in minimal media (*T*.test, $T_5 = -30$, $P < 0.001$, Holm adjusted for multiple testing, Fig. 3b). Deletion of *pms1* also resulted in significant cost (Rich: 13% fitness cost, *T*.test, $T_5 = -4.9$, $P < 0.05$, Minimal: 25% fitness cost, *T*.test, $T_5 = -15.5$, $P < 0.001$, Holm adjusted for multiple testing), although lower than *Δmsh2* (two sample *t*.test, $T_{9.68} = -4.8$, $P < 0.0001$). In contrast, the *msh6* null mutant did not have a significant decrease in relative fitness measured at transfer 5 when competed in either rich or minimal media (Fig. 3), however, the fraction of *Δmsh6* had reduced significantly by transfer 4 when competed in minimal media (Wilcoxon test, $W = 0$, $P < 0.01$). Thus, over longer periods of time there are significant costs associated with the loss of function of MutS or MutL complex.

**Msh6 G233A increases mutation rate and is correlated with increased mutation rates in clade A but not a fitness cost**

Since we find deletion of MMR components to be costly, as well as increasing rates of anti-fungal resistance, we asked whether the more subtle variants we see in our strain collection have a similar effect. As we did not have access to the rare clade B strains harbouring the *msh6-*

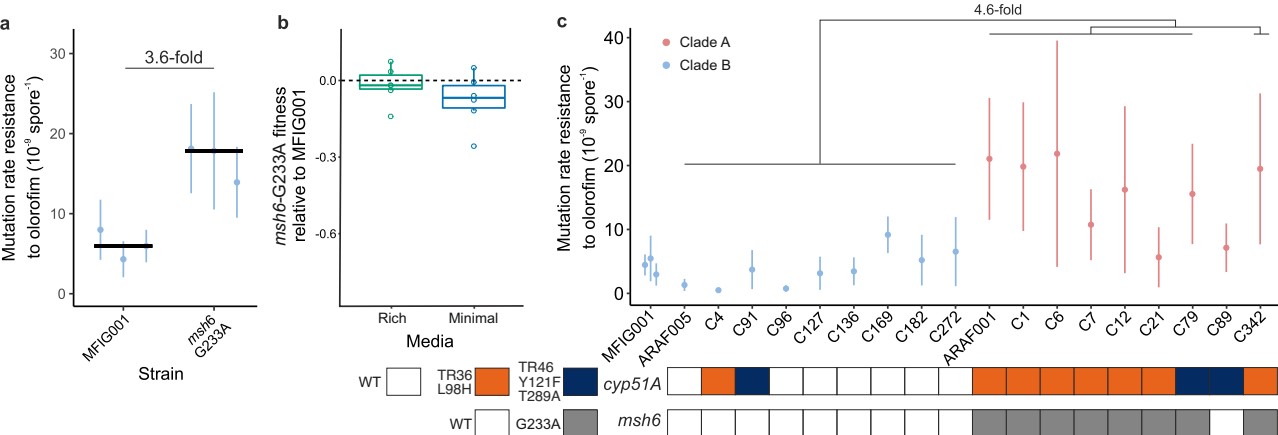

**Fig. 4 | *msh6*-G233A elevates mutation rates in neutral background and in natural isolates from clade A. a** Each point shows the calculated mutation rate from a single independent fluctuation test using 12 replicate cultures. Each of the three points for the MFIG001 *msh6*-G233A variant is a separate independent transformant. Error bars show 95% confidence intervals. Fold change shows the median fold change in mutation rate from the parental MFIG001 strain. **b** Mean fitness of *msh6*-G233A relative to MFIG001 across five transfers, the cross bar shows the median (three independent transformants, to replicates each for a total of $N = 6$), the lower and upper hinges correspond to the first and third quartiles and the whiskers extend to 1.5*IQR. Box plots coloured by media type. The horizontal dashed line shows equal fitness. **c** Mutation rate of natural genotypes to olorofim. Each point shows the calculated mutation rate from a single independent fluctuation test using 12 replicate cultures. Error bars show 95% confidence intervals. Fold difference shows the median fold change in mutation rate of isolates with G233A allele from clade B isolates. Points coloured by clade, red clade A, blue clade B. Key below plot shows the presence of azole resistance mutation *cyp51A* TR$_{34}$/L98H in orange, TR$_{46}$/Y121F/T289A in blue, and the presence of *msh6*-G233A in grey. Source data are provided as a Source Data file.

G233A variant, we reconstructed the G233A variant of *msh6* within MFIG001 (which clusters within clade B) through marker-less CRISPR-Cas9 mediated transformation[48] to determine the effect of the variant on mutation rate. The *msh6* variant resulted in a modest but significant increase in rates of resistance mutation to olorofim of 3.6-fold relative to the isogenic parental strain in three independent transformants (Fig. 4a, two sample ML-test, MFIG001 vs. *msh6*-G233A $T = -2.9576$, $P < 0.01$). A similar increase in mutation rate in the MFIG001 vs. *msh6*-G233A relative to its isogenic parental strain was observed when voriconazole was used as the selective marker (two sample ML-test, MFIG001 vs *msh6*-G233A $T = -3.2727$, $P < 0.01$, Fig. S8). Moreover, the increase in mutation rate was not associated with a significant increase in cost in either rich or minimal media (Fig. 4b, one sample $T$.test mu = 0, ACM $T_5 = -0.60839$, $P = 1$, AMM $T_5 = -1.8148$, $P = 0.51$, Holm adjusted for multiple testing). We, therefore, hypothesise that this variant is associated with variation in mutation rates in natural isolates. To test this, we assayed the rate of mutation to olorofim resistance in 18 isolates with a range of variants in *msh2*, *msh6* and *pms1*, with a combination of *cyp51A* resistance variants including wild-type, TR$_{34}$ and TR$_{46}$ from clade A and clade B (Table 1). Only the novel antifungal olorofim could be used for these fluctuation tests, as some isolates already had mutations in *cyp51A* that conferred resistance to azoles. The olorofim MICs of the isolates were not significantly different between isolates, and all fell at least 4-fold below the concentration used to select for resistant mutants (Fig. S9), enabling direct comparisons of mutation rates to be made. The isolates from clade B without variants in the MMR genes did not have significantly different mutation rates compared to MFIG001, including a clade B isolates with *cyp51A* TR$_{46}$/Y121F/T289A or TR$_{34}$/L98H azole resistance mutations (Fig. 4c, Two sample ML-test, MFIG001-clade B isolates, all $P > 0.05$). In contrast, isolates from clade A with the *msh6*-G233A variant had between a 1.3- and 5.1-fold increase in mutation rate (median 4.6-fold), significantly increasing the likelihood of spontaneous olorofim resistance arising (Fig. 4d, Two sample ML-test, MFIG001 vs *msh6*-G233A isolates, $P < 0.05$ in 7/8 strains). The mean difference in mutation rate to olorofim resistance between the *msh6*-G233A and *msh6*-WT populations was 1.22e−08 ($T$-test, $T_{8.82} = 5.76$, $P > 0.001$). These isolates encompassed both TR$_{34}$/L98H and TR$_{46}$/Y121F/T289 *cyp51A* resistance

genotypes and isolates with clinical and environmental origins (Table 1). However, two clade A isolates, C89 without *msh6*-G233A, and C21 with *msh6*-G233A, did not display similarly elevated mutation rates. Previous studies[44,45], together with Sanger sequencing 28 spontaneous olorofim-resistant mutants, show that olorofim resistance is mediated by a single point mutation in the drugs' target site; therefore, it is unlikely that the mutational target size for resistance is different between the two clades. Reverting the *msh6*-G233A allele to its WT form in a clade A isolate (C6) also resulted in a halving of mutation rate in three independent marker-less transformants (two sample ML-test, $P < 0.001$, Fig. S10). These results show that the presence of the G233A variant in *msh6*, unique to clade A, is associated with elevated rates of mutation to resist a novel antifungal in natural isolates. Moreover, this elevated mutation rate was not influenced by the genotype of the linked azole resistance locus.

## Discussion

The widespread use of agricultural azoles in the environment has been selected for cross-resistance to critical clinical first-line therapies used against *A. fumigatus* infections[18,20,49]. As a result, a fungicide-driven multi-azole-resistant clade has arisen, typified by the expansion of the TR$_{34}$/L98H allele in *cyp51A*. Our results show that isolates from this clade are not only resistant to azoles but also have elevated mutation rates due to a linked variant allele of *msh6*, a key protein in the MMR system involved in the recognition of base-base mispairings. It is unlikely that the *msh6* variant allele is causative of TR$_{34}$/L98H mediated azole resistance within this clade as *msh6* is predominantly responsible for recognising and repairing single base mismatches, and tandem repeats were never observed in the spontaneous resistant mutations isolated from the fluctuation test. The two alleles likely arose together due to the *msh6* allele hitchhiking on strong selective sweeps for azole resistance. However, this modest but significant increase in mutation rate in clade A isolates increases the likelihood of resistance emerging to other antifungals in this already azole-resistant clade.

*A. fumigatus* produces vast numbers of spores, and due to their mononucleate nature, even slightly elevated rates of mutation, such as those measured here, mean it is highly likely that in any given niche,

**Table 1 | Natural isolates tested for changes in mutation rate**

| Strain | Source | Sample | cyp51A | Clade | pms1 | msh2 | msh6 |
|---|---|---|---|---|---|---|---|
| ARAF005 | Clinical | Sputum | Wild-type | B | Wild-type | Wild-type | Wild-type |
| C4 | Environmental | Wheat field | TR$_{34}$/L98H | B | Wild-type | Wild-type | Wild-type |
| C91 | Environmental | Plant bulb | TR$_{46}$/Y121F/T289A | B | Wild-type | Wild-type | Wild-type |
| C96 | Environmental | Plant bulb | Wild-type | B | Wild-type | Wild-type | Wild-type |
| C127 | Clinical | Sputum | Wild-type | B | Wild-type | Wild-type | Wild-type |
| C136 | Clinical | Sputum | Wild-type | B | Wild-type | Wild-type | Wild-type |
| C169 | Clinical | Sputum | Wild-type | B | Wild-type | Wild-type | Wild-type |
| C182 | Clinical | Sputum | Wild-type | B | Wild-type | Wild-type | Wild-type |
| C272 | Environmental | Unknown | Wild-type | B | M315V | Wild-type | Wild-type |
| ARAF001 | Clinical | Sputum | TR$_{34}$/L98H | A | E444G/S758Y | Wild-type | G233A |
| C1 | Environmental | Soil | TR$_{34}$/L98H | A | Wild-type | Wild-type | G233A |
| C6 | Environmental | Soil | TR$_{34}$/L98H | A | E444G | Wild-type | G233A |
| C7 | Environmental | Soil | TR$_{34}$/L98H | A | Wild-type | Wild-type | G233A |
| C12 | Environmental | Soil | TR$_{34}$/L98H | A | Wild-type | Wild-type | G233A |
| C21 | Environmental | Soil | TR$_{34}$/L98H | A | Wild-type | Wild-type | G233A |
| C79 | Environmental | Compost | TR$_{46}$/Y121F/T289A | A | Wild-type | K816Q | G233A/A1090S |
| C89 | Environmental | Plant bulb | TR$_{46}$/Y121F/T289A | A | Wild-type | E812G | Wild-type |
| C342 | Environmental | Soil | TR$_{34}$/L98H | A | E444G | Wild-type | G233A |

where millions to billions of spores are produced, at least one spontaneous resistant mutant will arise. Although mutational instability is associated with significant fitness costs in stable environments it is telling that deletion of msh6 resulted in only a minor reduction in fitness costs in vitro, however these costs will be dependent upon the environmental conditions in which they are measured. Though fitness costs may accumulate over longer evolutionary time periods, the relatively mild medium-term reduction in fitness that we observe in strains containing msh6-G233A might explain why the msh6-G233A allele has persisted in A. fumigatus populations exposed to azole fungicides since their introduction in the mid-1970s[50,51]. The complex lifecycle of A. fumigatus, which can undergo both asexual and sexual reproduction[52], has also likely influenced the maintenance of the mutator phenotype[53]. The recombination rate of A. fumigatus is recorded to be the highest of any eukaryotic organism, with a recombination rate of 0.422 cM/kb, producing an average of 29.9 crossovers per chromosome[54]. This would result in at least one crossover event within the 360 kb window between cyp51A and msh6 for each sexual cross. However, the frequency of sexual crossing in the environment is currently unknown. A high recombination rate between isolates within clade A may act to dissociate the mutator allele from any deleterious mutations it may cause, thus decreasing any selective disadvantage of the mutator allele[21,53,54] while reproductive isolation between clades A and B will act to maintain the linkage of TR$_{34}$/L98H and msh6-G233A given that almost every strain within clade A have both alleles. Additionally, it would be expected that purifying selection would remove deleterious mutations from the population. This apparent reproductive isolation and purifying selection against costly mutations may help to explain the strong genetic linkage between clade A, TR$_{34}$/L98H and the msh6 mutator allele, its ongoing environmental persistence and would contribute to the maintenance of the mutator phenotype within this clade.

Environmental azole resistance is typified by two mutations, a tandem repeat in the promoter and a point mutation in the target site cyp51A. Given that Msh6 acts predominantly upon base-pair mismatches, and we did not observe tandem repeats in our spontaneous resistant mutants, we do not hypothesise that the mutator allele has given rise to this canonical mechanism of azole resistance. While we cannot state that the two are causally linked due to the close genetic linkage between cyp51A and msh6, the mutator allele may be hitchhiking on strong selection for the azole-resistant background. This linkage may also help to explain the accumulation of multiple independent resistance mechanisms to different classes of agricultural antifungals within the mutable clade[22]. In addition, our results show diversity in mutation rates across the A. fumigatus phylogeny, including within clade A (Fig. 4c), suggesting that the genetic context in which the msh6 variant allele is found may influence its impact upon mutation rate. The use of agricultural azoles has led to selective sweeps across multiple loci in clade A resulting in significant differences between the genomic backgrounds in clade B and clade A. As a result, other genetic factors may also play a role in the observed differences in mutation rate between clade A and B. Missense mutations in other MMR genes, particularly msh2 and pms1, may also be contributing to the variance in mutation rate observed within clade A. Notably, three of the clade A strains tested also had the pms1 variant E444G (ARAF001, C6 and C342), and these three strains also had the highest mutation rates. However, these mutations are rare within the population. Nonetheless, the strong association of the mutator phenotype with azole resistance may mean that we are witnessing the early stages of the anthropogenic evolution of a pathogenic species that can not only more rapidly adapt to antifungal challenge, but also other environmental challenges encountered by this fungus, such as climate change.

Mutations leading to hypermutator phenotypes are a common adaptive trait within prokaryotic populations[55], leading to vastly elevated mutation rates, and are often associated with chronic infections where strong fluctuating selection occurs[46]. In the short term, elevated mutation rates can be a source of beneficial mutations[56] but are considered detrimental due to the accumulation of damaging mutations and are ultimately selected against in the long term. Nonsense mutations in Msh2 within an outbreak of C. deuterogatti have previously been demonstrated to be coupled with elevated mutation rates and the emergence of antifungal resistance[29]. Here, we show that rather than being constrained to an isolated outbreak or resulting in very large increases in mutation rate, a globally distributed cluster of genotypes has a significant but moderate increase in mutation rate due to a variant allele in msh6. This variant is stable across niches and is transmitting[20], having been isolated within multiple different environmental and clinical settings, and through time, with isolates sampled over a 12-year period (Supplemental dataset 1). This relatively modest

increase in mutation rate appears to avoid the deleterious consequences of more severe hypermutator phenotypes, displaying no significant fitness cost. We do not observe increased phylogenetic branch lengths associated with azole-resistant isolates containing the *msh6* variant allele, potentially due to an ancestral selective sweep due to the recent expansive use of azoles. Isolates within clade A, which have the *msh6*-G233A allele on average, have 3.3% more SNPs than clade A isolates without the MMR variant, but this difference is not significant (*T*-test, $T_{20.7} = -1.26$, $P = 0.221$). However, recent mathematical modelling predicts that although subtle changes in mutation rates can play important roles in the long-term success of genetic variants, such differences will be difficult to detect using traditional metrics such as substitution rates[57]. The drift-barrier hypothesis predicts that mutation rates rise with reduced effective population sizes[53,58], therefore a potentially recent population bottleneck through the emergence of the reproductively isolated Clade A may have increased effect of genetic drift, ultimately resulting in diminished benefits of optimised mismatch repair. However, we currently have a poor understanding of the effective population size of *A. fumigatus*, particularly in relation to the specific niches where selection may be occurring, which will require in-depth global environmental surveillance.

Strong fluctuating selection through the extensive use of multiple different classes of fungicide may have provided strains with elevated mutation rates a fitness advantage by more rapid adaptation to changing abiotic stresses. Alternatively, as the *msh6* allele is closely linked to *cyp51A* on chromosome 4, it may be hitchhiking on strong directional selection for the TR$_{34}$/L98H azole resistance mutation. Nevertheless, isolates from the multi-azole-resistant clade A are between two- and five-fold more likely to acquire de novo mutations that provide resistance to novel antifungal therapies. Moreover, the impact of this elevation in mutation rate could be important in the within-patient evolution of antifungal resistance leading to more rapid treatment failure for CPA patients on long-term antifungal therapy due to increased mutational supply. The advent of combination therapy using novel, orally bioavailable classes of antifungal drugs with new modes of action including fosmanogepix (Phase II clinical trials) and olorofim (Phase III) should reduce the impact of antifungal resistance that arises in the patient. However, a number of agricultural antifungals have either been approved or are in late development for use in crop protection that share the same mechanism of action as these novel clinical drugs[59]. The pyridine fungicide aminopyrifen[60] targets the same enzyme, GWT-1, as fosmanogepix. Likewise, ipflufenoquin, which has been approved by the US Environmental Protection Agency for agricultural use[61], is analogous to the dihydroorotate dehydrogenase (DHODH) inhibitor olorofim. Further, ipflufenoquin has been shown to select for cross-resistance to olorofim[45]. The dual use of these novel classes of antifungal drugs coupled with increased mutation rates in azole-resistant isolates increases the probability of the accumulation of multiple independent resistance mechanisms within clade A to both azole and novel antifungal compounds. Through the pervasive anthropogenic use of agricultural azoles, a lineage of *A. fumigatus* has been selected that is multi-azole resistant, multi-fungicide resistant, and also displays increased adaptability, which ultimately will lead to the evolution of a lineage that manifests pandrug resistance.

## Methods

### Strains, culture conditions and antifungals

*A. fumigatus* MFIG001 was used as the parental strain to generate *msh2*, *msh6*, and *pms1* deletion strains and *msh6*-G233A mutant by CRISPR-Cas9 mediated transformation[48]. *A. fumigatus* clinical and environmental isolates were provided by Matthew Fisher[20] and are described in Table 1. *A. fumigatus* strains used in this study were cultured on Aspergillus Complete Media (ACM)[62] agar for 3 days at 37 °C unless stated otherwise. Spores were harvested in phosphate-buffered saline (PBS) + 0.1% Tween-20 and filtered through miracloth (Millipore). Olorofim was synthesised by Concept Life Sciences, voriconazole was obtained from Merck (32483) and itraconazole from Cayman Chemical (13288). See Supplementary dataset 3 for primer sequences used in this study.

### Gene deletion and CRISPR-Cas9 gene modification

To construct the deletion mutants of MFIG001, mismatch repair genes *msh2*, *msh6* or *pms1* were replaced with the hygromycin resistance cassette. Target-specific crRNAs and associated PAM sites flanking each gene were designed using EuPaGDT[63] based on the A1163 genome sequence. Homology-directed repair (HDR) templates, including the hygromycin resistance cassette, were amplified from the pAN7.1 plasmid using primers that incorporated 50-bp microhomology arms flanking the gene to be replaced. CRISPR-Cas9 transformation was conducted for *A. fumigatus* as follows[48]; MFIG001 conidia were inoculated into liquid ACM and cultured for 16 h at 37 °C, shaking at 120 rpm. Mycelia were harvested by filtration through miracloth followed by protoplasting using Vinotaste (Lamoth-Abiet) 10 g/mL in 0.6 M KCl + Citric acid at 37 °C for 4 h, shaking at 120 rpm. Protoplasts were harvested and centrifuged at 1800×*g* for 10 min, followed by washing three times with 0.6 M KCl. Protoplasts were resuspended in 0.6 M KCl + 200 mM CaCl$_2$ and diluted to $1 \times 10^6$ protoplasts/mL. RNP complexes were assembled using Alt-R *S.p.* Cas9 Nuclease V3, Alt-R® CRISPR-Cas9 tracrRNA and locus-specific Alt-R® CRISPR crRNA (Integrated DNA Technologies) by heating to 95 °C for 5 min and gradually cooling to room temperature in a thermal cycler for 10 min. HDR template (1 μg), RNPs (33 μM RNA duplex with 1.5 μg Cas9) and protoplasts were mixed with PEG-CaCl$_2$ (60 wt%/vol PEG3350, 50 mM CaCl$_2$, 450 mM Tris−HCl, pH 7.5) and incubated on ice for 50 min. Protoplasts were then spread onto YPS plates containing 100 μg/mL hygromycin (TOKU-E). Plates were incubated for 24 h at room temperature, followed by incubation at 37 °C for 3 days. Transformants were purified and validated by PCR using primers flanking the loci (Supplementary dataset 3), coupled with primers internal to the hygromycin resistance cassette, as well as primers specific to the loci deleted. For selection-free CRISPR-Cas9 to recreate *msh6*-G233A in MFIG001 and version of *msh6*-G233A in C6, two single-stranded repair templates were designed, which incorporated the G698C or C698G transversion into MFIG001 and C6 respectively. Transformation was conducted as above and transformed protoplasts were plated onto non-selective YPS plates. MFIG001 colonies were screened for *msh6* c.G698C and C6 screed for c.C698G by PCR using SNP-specific primers (Supplementary dataset 3), and positive transformants were confirmed by Sanger sequencing *msh6* (Genewiz). Three independent transformants from three separate transformations were selected for mutation rate and fitness analysis in each background.

### Antifungal susceptibility testing

MIC determination for all drugs was carried out according to the broth microdilution method outlined by the European Committee on Antimicrobial Susceptibility Testing[64]. $1 \times 10^5$ spores/ml were inoculated per well in CytoOne® 96-well plates (StarLab) containing RPMI-1640 medium supplemented with 2.0% glucose and MOPS pH 7.0. Growth was measured by OD at 600 nm in a two-fold dilution series of antifungal drugs (olorofim 0.00195-2 μg/ml, voriconazole 0.0156−16 μg/ml) following incubation at 37 °C for 48 h. Three biological replicates for each strain tested were conducted. The MIC was defined as the lowest concentration required to reduce growth by ≥90% relative to the drug-free control wells as defined by the EUCAST method for susceptibility testing of moulds (version 9.4).

### Fluctuation test and mutation rate analysis

Fluctuation tests were performed by inoculating 12 independent 10 ml cultures with 5000 spores enumerated by hemacytometer on ACM agar

from a single spore stock and allowed to grow in non-selective conditions for 72 h at 37 °C. To determine the effect of the mutagen ethyl methanesulfonate on mutation rate of MFIG001, a two-fold series of EMS, between 64 and 512 μg/ml, was added to the growth media. Spores were harvested in PBS + 0.1% tween-20, spores were then pelleted by centrifugation and resuspended in 500 μl PBS + 0.1% tween-20. 2 μl of spores from each culture were removed, diluted, and used to measure the final population size ($N_t$) by flow cytometry. We obtained the number of spontaneous resistant mutants that formed during growth in non-selective conditions by transferring the total remaining spores to ACM agar plates containing either 2 μg/ml voriconazole, 8 μg/ml itraconazole or 0.5 μg/ml olorofim. Plates were incubated for 72 h at 37 °C. For Δmsh2, Δmsh6 and Δpms1 (Fig. 2b) and msh6-G233A (Fig. 4a, b) mutation rates against voriconazole and olorofim were calculated. An estimate of the number of mutation events ($m$) was calculated from the distribution of the number of mutants across the 12 independent cultures using the Ma–Sandri–Sarkar maximum-likelihood method implemented in the *flan* R package. To calculate the mutation rate per spore $m$ was divided by the final population size $N_t$. For Δmsh2, Δmsh6 and Δpms1 mutation frequency to itraconazole and olorofim (Fig. 2c, d) were calculated from six independent cultures by dividing the number of resistant colonies by the number of spores plated. For each fluctuation test, at least three randomly selected spontaneous resistant mutants were sub-cultured on non-selective ACM for a further 72 h at 37 °C followed by culturing on ACM containing 2 μg/ml voriconazole, 8 μg/ml itraconazole or 0.5 μg/ml olorofim to confirm resistance. The genes encoding targets of voriconazole, *cyp51A*, and olorofim, *pyrE*, of 27 voriconazole and 27 olorofim spontaneous resistance mutants were also Sanger sequenced.

## Fitness assays

Radial growth rates of Δmsh2, Δmsh6 and Δpms1 mutants were conducted by spotting $10^3$ spores per strain on ACM or Aspergillus Minimal Media (AMM) agar plates. Plates were incubated for 96 h and the radius of the colonies was measured every 24 h. Three biological replicates were conducted for each strain and media condition. To measure fitness over longer time scales, culture flasks containing 10 ml of ACM or AMM were inoculated with 1:1 mixture of MFIG001 plus either Δmsh2, Δmsh6 and Δpms1 mutant strains at a final density of 5000 spores/ml. The initial ratio of WT to mutant was determined by diluting the spore mixture to the appropriate concentration and plating it onto non-selective and hygromycin plates to calculate the total population size and fraction of deletion mutant in the mixture. The cultures were incubated for 72 h at 37 °C at which point total spores were harvested, 1% of the total harvested spores were used to inoculate fresh ACM or AMM. The remaining was diluted and plated on non-selective and hygromycin plates to calculate the frequency of deletion mutants within the population over time. This was repeated for a total of 5 transfers. Equal fitness would result in the maintenance of a 1:1 ratio throughout the experiment. The same method was conducted to measure the fitness cost of msh6-G233A, however MFIG001 transformed with a hygromycin resistance cassette (*hph*) at the neutral insertion site *aft4*[65] was used as the competitor. Fitness of the *hph* cassette at *aft4* was accounted for by competition against wild-type MFIG001. A total of six replicates were conducted, two for each of the three independent msh6-G233A transformants.

## Bioinformatics

Analysis of variants in MMR genes was conducted on 218 previously sequenced *A. fumigatus* isolates (Supplemental dataset 1, ENA:PRJEB27135)[20]. Orthologous sequences of mismatch repair genes *msh2* (AFUB_039320, AFUA_3G09850), *msh3* (AFUB_090020, AFUA_7G04480), *msh6* (AFUB_065410, AFUA_4G08300), *pms1* (AFUB_029050, AFUA_2G13410) and *mlh1* (AFUB_059270, AFUA_5G11700) were extracted from the previously published pan-

genome[20] by identifying the representative pan-genes using BLASTP v2.12.0. Extracted sequences underwent multiple-sequence alignment using MUSCLE v3.8.1551[66] and variants were identified using SNP-sites v2.5.1[67] and confirmed by visualisation in JalView v2.11.2.6[68]. Whole-genomic single nucleotide polymorphism (SNP) phylogeny of UK isolates was produced as described in Rhodes et al.[20]. Phylogenetic trees with overlaying metadata were generated using iTOL v6.5.4[69]. Per-site $F_{ST}$, the measure of population differentiation due to genetic structure, was calculated for Clade A vs Clade B. It was assumed that isolates with the *msh6* variant would mostly be found in Clade A, using VCFtools v0.1.13[70].

For analysis of global isolates, publicly available raw reads (Source data file) were mapped to *Af293* (GCF_000002655.1) using Burrows-Wheeler Aligner v0.7.17[71]. Text pileup outputs were generated for each sequence using SAMtools v1.6 mpileup with option –I to exclude insertions and deletions[72]. BCFtools v1.6 call was used to call SNPs with options -c to use the original calling method and --ploidy 1 for haploid data[73]. Consensus genome sequences in fasta format were extracted from vcf files using seqtk v1.2 (available at https://github.com/lh3/seqtk [github.com]) with bases with a phred quality score below 40 counted as missing data. MEGA version X[74] was used to create a neighbour-joining tree using the Tamura-Nei model with 100 bootstraps for all whole genome sequences. The phylogeny was visualised using iTOL[69].

Sequence conservation at the G233 locus of *msh6* was examined in 289 members of the family Trichocomaceae retrieved from the NCBI non-redundant protein database through protein blast using *Af293* Msh6 (AFUA_4G08300) as the reference. Sequences were aligned using MUSCLE v3.8.1551 and sequence logos were created using WebLogo[75].

## Whole genome sequencing of spontaneous resistant mutants and variant calling

Six independent, spontaneous voriconazole-resistant mutants from six different fluctuation assay plates were selected for whole genome sequencing generated from an MMR deletion background (Δmsh2, Δmsh6 and Δpms1) and in the wild-type MFIG001. In parallel, the parental voriconazole sensitive MFIG001, Δmsh2, Δmsh6 and Δpms1 were also whole genome sequenced. Total gDNA was extracted from conidia using the CTAB DNA extraction protocol, the integrity of the DNA was assessed on a 0.75% agarose gel and quantified by Qubit using the dsDNA HS Assay Kit (Thermo Fisher Scientific). Library preparation and sequenced of total genomic DNA was conducted by Earlham Institute (http://www.earlham.ac.uk), using Nextera DNA Flex library preparation protocol and 150 base pair paired-end reads on the Illumnia NovaSeq 6000 platform. Read quality was assessed using FastQC[76], and sequence adapters were trimmed using Trimmomatic v0.39[77]. De novo assembled genomes were produced for each of the parental voriconazole-sensitive strains using SPAdes v3.15.4[78]. Low complexity regions were masked using RepeatMasker v.4.1.2-p1[79], and gene annotations were generated through a BLASTn v2.9.0[80] search using A1163 reference genes obtained from FungiDB (release 65)[81], against the masked genomes. Trimmed reads from the spontaneous voriconazole-resistant mutants were aligned to the appropriate parental de novo assembly using the Burrows–Wheeler Aligner MEM v0.7.17-r1188[82] and converted to sorted BAM format using SAMtools v.1.3.1[72]. Variant calling was conducted using GATK HaplotypeCaller 4.1.8.0[83] and low confidence calls were filtered using VariantFiltration (DP < 10, RMSMappingQuality < 40.0, QualByDepth < 2.0, FisherStrand > 60.0, ABHom < 0.9). Binary presence/absence of variants in each strain were created using SAMtools "bcftools" and variants were manually confirmed using igv-reports v1.11.0[84,85].

## Statistics

All statistics were conducted in R v4.1.1. Assumptions of normality were determined using Q–Q plots and Shapiro–Wilk tests, and where

appropriate non-parametric tests were conducted. Pearson's chi-squared tests were used to test for significant differences between expected and observed frequencies of variant alleles between clade A and clade B and between TR$_{34}$/L98H and WT *cyp51A* genotypes. Significance differences between mutation rates were calculated using maximum likelihood estimation using the Luria-Delbrück model (*flan* v0.9 package R). Kruskal–Wallis test tested for significant differences in MIC of spontaneous resistant mutants in *Δmsh2*, *Δmsh6*, and *Δpms1* backgrounds. Pairwise differences between groups were determined by ANOVA with Tukey post hoc tests where assumptions were met and Wilcoxon rank sum tests otherwise. To model the effect of transfer, strain and media upon fitness a mixed effects linear model was fitted to the data using the *nlme* v3.1-166 package in R. The model fits the frequency of mutant strain to the interacting effects of transfer, strain, and media type, with replication modelled as a random effect. The model allowed variance to change with fitted values to account for the reduced variance as mutant frequency approached zero. The fitness at each timepoint was calculated as

$$\text{Fitness} = \log\left(\frac{\left(\frac{F_{KO_e}}{F_{KO_i}}\right)}{\left(\frac{F_{WT_e}}{F_{WT_i}}\right)}\right)$$

where $F$ is the fraction of either deletion strain (KO) or wild-type strain (WT) of the inoculum (i) or end (e) of each growth cycle. The mean fitness across all five transfers is reported in Fig. 3b. Significant difference from equal fitness of 0 was calculated using a two-sided t-test, using Holm correction for multiple testing.

### Reporting summary

Further information on research design is available in the Nature Portfolio Reporting Summary linked to this article.

## Data availability

The sequencing data generated in this study have been deposited in the European Nucleotide Archive database under accession no. PRJEB81974. Whole genome sequence data of UK isolates was conducted as parot of Rhodes et al.[20], with reads deposited under accession no. PRJEB27135. All other data generated in this study are provided in the Supplementary Information/Source Data file. Source data are provided with this paper.

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

## Acknowledgements

This work was funded by the Wellcome Trust under project no. 221653/Z/20/Z (awarded to M.J.Bo.), Wellcome Trust grant no. 219551/Z/19/Z (to J.L.R., M.C.F. and M.J.Br.), Wellcome Trust grant 208396/Z/17/Z (to M.J.Br.), and by the UK Medical Research Council (MRC) no. MR/R015600/1 (to M.C.F.). H.C. received a Doctoral Training Programme (DTP) studentship from the Biotechnology and Biological Sciences Research Council (BBSRC). N.v.R. was supported by Wellcome Trust grant no. 226408/Z/22/Z. This work was also supported by the Centers for Disease Control and Prevention (CDC; contract 0HCVLD13-2018-27470 to M.M. and M.T.B.) and United States Department of Agriculture, National Institute of Food and Agriculture (USDA NIFA AFRI grant 2019-67017-29113 to M.T.B. and M.M.). B.N.C.-S. was supported by the National Science Foundation under grant no. DGE-1545433.

## Author contributions

Conceptualisation: M.J.Bo. and M.J.Br. Methodology: M.J.Bo., N.v.R., H.C., J.L.R., C.G.K., and M.J.Br. Statistics: M.J.Bo., H.C., and J.L.R. Formal analysis: M.J.Bo., B.N.C-S., and J.L.R. Investigation: M.J.Bo., N.v.R., H.C., J.L.R., and B.N.C.-S. Data curation: M.J.Bo., and B.N.C.-S. Writing—original draft: M.J.Bo. and M.J.Br. Writing—review and editing: all authors. Visualisation: M.J.Bo. and B.N.C.-S. Supervision: M.J.Bo., C.G.K., M.T.B, M.M., M.C.F, and M.J.Br. Funding acquisition: M.J.Bo., M.M., M.T.B., M.C.F., C.G.K., and M.J.Br.

## Competing interests

M.J.Br. is a former employee of F2G Ltd (to 2008) and has received funding from F2G Ltd for a Ph.D. studentship. The remaining authors declare no competing interests.
