## [Transparent Peer Review file · Nature Communications]

Elevated mutation rates in multi-azole resistant *Aspergillus fumigatus* drive rapid evolution of antifungal resistance

Corresponding Author: Dr Michael Bottery

Version 0:

Reviewer comments:

Reviewer #1

(Remarks to the Author)

I have reviewed this manuscript previously. I find that the authors have satisfactorily addressed my comments, and I recommend acceptance. My final suggestion is to fix the title by replacing "drives" with "drive" (because the subject is "rates", plural).

Reviewer #2

(Remarks to the Author)

I feel that the authors have made a strong effort to address the points raised experimentally, or otherwise provided a rationale that addresses them.

There might be some scope for some minor editing to tidy up the text. For example:

msh6 in italics or not? Lines 54, 388, 514 probably in italics? A mix of msh6::G233A and msh6-G233A. On page 13, if talking about an allele that would be a gene, so msh6 in italics.

Wild-type vs. wildtype.

Line 170: cyp51A.

Line 214: MFIG0001.

Hence, probably worth a final read over the manuscript for minor things like these.

Reviewer #3

(Remarks to the Author)

I have reviewed this article for the second time, and I acknowledge the effort made by the authors in performing additional experiments suggested by the reviewers and in clarifying some of the aspects regarding the interpretation of the results. I think the article is now more clear and to the point and the claims are more cautious and well explained. Although how the association between these two mutations first appeared is unclear, the authors hypothesis is plausible in light of the data, and the increased likelihood of these combined strains to adapt to new generation drugs such as olorofilm is clear. I only have two remaining comments, but none of them is critical.

Comments.

-Line 200. It is remarkable that 37% (10/27) of the randomly selected spontaneously resistant isolates had acquired the exact

same known voriconazole resistance allelic variant G448S. How unexpected is this?. Not being an expert on *A. fumigatus*, I wonder whether other resistance variants are known in this gene, and if that is the case, why they were never observed in this experiment. This points to either a much stronger effect of this mutation, a deleterious effect or others, or a mutational bias. In this regard are all of these mutations involving the same mutation at the nucleotide level?

- line 226- How can an elevated mutation rate result in the purging of deleterious mutations?, it is selection (not mutation) that purge deleterious mutations. What I think the authors mean here is that many of the mutations originated in this mutants were deleterious and therefore not observed. This again points to a different pattern of mutations generated by the different mutants, likely related to the different functions of the genes. Alternative explanations are possible, for instance, some repair mechanisms are linked to gene expression, which may reduce mutations in coding regions with respect to intergenic regions.

Reviewer #4

(Remarks to the Author)

Resistance to the azole antifungals in the pathogenic mold *Aspergillus fumigatus* has emerged as a significant threat to human health. The azoles voriconazole and isavuconazole are front line therapy for life-threatening invasive infections due to this organism. Several resistance determinants have been identified, most commonly mutations in *cyp51A* which encodes the azole target sterol demethylase. Extensive use of agricultural azoles has led to resistance in the environment and subsequent infections due to resistant isolates. This includes a widespread multi-azole resistant genetic cluster of genotypes (clade A). The authors demonstrate here that the G233A variant in the MutS component gene *msh6* is strongly associated with the TR34/L98H allele of *cyp51A* in clade A. They go on to show that while both MutA and MutL deletion mutants exhibit a hypermutator phenotype and that defects in DNA mismatch repair result in costs to fitness, the G233A variant in *msh6* increases mutation rate, correlates with increased mutation rates in clade A isolates but does not confer a fitness cost. Importantly, while this observed in the context of azole resistance, the greater mutation rate of these isolates appears to have emerged concomitantly and not as a contributor to the azole resistance mutations in these isolates, and results in a greater propensity to evolve resistance to new antifungals.

The revised manuscript is well-written and clearly presents the problem and hypotheses explored in this body of work. Results are likewise well organized and clearly presented with effective use of figures and tables. Methods are appropriate and easily understood as presented. In the discussion, the authors note the important point that since Msh6 is responsible for dealing with single base mismatches, it is not likely that mutations in this gene are responsible for the tandem repeats of the TR34/L98 azole resistance *cyp51A* allele in these isolates but rather the two mutations more likely arose together. They conclude by further underscoring this multi-azole and multi-fungicide resistant and highly genetically adaptable lineage of *A. fumigatus* is the product of pervasive use of agricultural azoles. The authors have been extensively responsive to all reviewers' comments with added experimentation where requested. All of my previous questions and concerns have been satisfactorily addressed.

Open Access This Peer Review File is licensed under a Creative Commons Attribution 4.0 International License, which permits use, sharing, adaptation, distribution and reproduction in any medium or format, as long as you give appropriate credit to the original author(s) and the source, provide a link to the Creative Commons license, and indicate if changes were

made.

REVIEWERS' COMMENTS

Reviewer #1 (Remarks to the Author):

My final suggestion is to fix the title by replacing "drives" with "drive" (because the subject is "rates", plural).

RESPONSE: We agree with the reviewer's suggestion and have made this change.

Reviewer #2 (Remarks to the Author):

msh6 in italics or not? Lines 54, 388, 514 probably in italics? A mix of msh6::G233A and msh6-G233A. On page 13, if talking about an allele that would be a gene, so msh6 in italics.

Wild-type vs. wildtype.

Line 170: cyp51A.

Line 214: MFIG001.

RESPONSE: As suggested, we have read through and edited the manuscript to ensure consistence of notation.

Reviewer #3 (Remarks to the Author):

-Line 200. It is remarkable that 37% (10/27) of the randomly selected spontaneously resistant isolates had acquired the exact same known voriconazole resistance allelic variant G448S. How unexpected is this?. Not being an expert on *A. fumigatus*, I wonder whether other resistance variants are known in this gene, and if that is the case, why they were never observed in this experiment. This points to either a much stronger effect of this mutation, a deleterious effect or others, or a mutational bias. In this regard are all of these mutations involving the same mutation at the nucleotide level?

RESPONSE: Although there are several Cyp51A mutations that provide resistance to azoles, previous research indicates that the mutational spectrum of azole resistance appears to be in part dependent upon exposure to specific azoles. As suggested by the author, itraconazole resistance mutations such as M220I provide much lower effect under voriconazole treatment. G448S provides high level resistance to voriconazole and permits growth at the concentration used to select for spontaneous resistant mutants (2 µg/ml voriconazole), therefore these mutations are not unexpected. Perhaps, more unexpectedly were the non-cCp51a mutations that provided resistance to voriconazole, whole genome sequencing of spontaneous resistant mutants identified previously reported HMG-CoA azole resistance mutations but also parallel mutations across independent strains within coding regions not

previously reported to be associated with resistance. We leave the investigation into these for future work.

- line 226- How can an elevated mutation rate result in the purging of deleterious mutations?, it is selection (not mutation) that purge deleterious mutations. What I think the authors mean here is that many of the mutations originated in this mutants were deleterious and therefore not observed. This again points to a different pattern of mutations generated by the different mutants, likely related to the different functions of the genes. Alternative explanations are possible, for instance, some repair mechanisms are linked to gene expression, which may reduce mutations in condig regions with respect to intergenic regions.

RESPONSE: We agree that this sentence was misleading and lost the message that we were trying to convey. Higher mutation rates do not lead to purging of deleterious mutations, rather, as the reviewer points out, it is selection that purges the mutations. We have rephrase the sentence to address this point. The sentence now reads "...suggesting that the elevated mutation rates within these strains result in deleterious or lethal intragenic mutations which are purged by negative selection".